# Key Roles of Crypto-Exchanges in Generating Arbitrage Opportunities

**DOI:** 10.3390/e23040455

**Published:** 2021-04-12

**Authors:** Audrius Kabašinskas, Kristina Šutienė

**Affiliations:** Department of Mathematical Modelling, Kaunas University of Technology, Studentu 50, 51368 Kaunas, Lithuania; kristina.sutiene@ktu.lt

**Keywords:** arbitrage, crypto-currency, network, crypto-exchange, graph theory, centrality, canonical correlation

## Abstract

The evolving crypto-currency market is seen as dynamic, segmented, and inefficient, coupled with a lack of regulatory oversight, which together becomes conducive to observing the arbitrage. In this context, a crypto-network is designed using bid/ask data among 20 crypto-exchanges over a 2-year period. The graph theory technique is employed to describe the network and, more importantly, to determine the key roles of crypto-exchanges in generating arbitrage opportunities by estimating relevant network centrality measures. Based on the proposed arbitrage ratio, Gatecoin, Coinfloor, and Bitsane are estimated as the best exchanges to initiate arbitrage, while EXMO and DSX are the best places to close it. Furthermore, by means of canonical correlation analysis, we revealed that higher volatility and the decreasing price of dominating crypto-currencies and CRIX index signal bring about a more likely arbitrage appearance in the market. The findings of research include pre-tax and after-tax arbitrage opportunities.

## 1. Introduction

Arbitrage, being a core concept in finance, defines nearly the simultaneous sell and purchase of identical or similar financial securities in order to profit from price discrepancies in different markets. The concept of arbitrage is closely related but opposite to the theory of market efficiency, which defines the market as perfectly efficient when all equivalent assets converge to the same price. Many important findings of conventional financial economics are based on the assumption of no arbitrage or Law of One Price, and it serves as one of the most fundamental unifying principles for studying traditional financial markets and asset pricing [1,2]. However, researchers have determined a strong evidence to assert the opposite. In fact, it was found that there exist situations where arbitrage opportunities are observed across a range of financial instruments and do not quickly disappear. Moreover, in response to the limitations faced by a conventional financial economic theory, behavioral finance as a new approach to financial markets has emerged [3,4]. Consequently, some trading algorithms have been developed, which exploit the arbitrage opportunities in a short time period in order to benefit from automated trade execution, taking into account timing, risk, and transaction costs.

With the appearance of an entirely new digital asset class, namely a crypto-currency market, many different studies have been carried out to analyze this market focusing on various statistical characteristics [5,6,7], the relationship between crypto-currencies and other assets [8,9,10], regime switches [11,12,13], technological aspects [14,15], etc. Among them, only few recent papers published can be found that specifically explore the trading and formation of arbitrage in a crypto-currency market. For instance, the authors of study [16] present their approach to investigate the opportunities of arbitrage by making transactions between crypto-exchanges. First, they draw our attention to the main factors that lead to the price discrepancies and arbitrage formation in the crypto-space. Unlike the traditional regulated markets, the crypto-currency market lacks of a centralized exchange and, moreover, suggests no provision to guarantee that investors get the best price when executing trades. Additionally, they reported that there exist periods of large, recurrent arbitrage opportunities across exchanges, using tick data for 34 exchanges across 19 countries. Moreover, the authors highlighted that the price differences are much larger for exchanges across ountries than within the same country, and smaller between crypto-currencies. Comparatively, a recent study [17] addressed the arbitrage opportunities on trading bitcoin at four crypto-exchanges. It was determined that deviations from price parity are much higher on average, more volatile, exhibit persistency, and occasionally achieve substantially large extremes during the 2016–2017 period. Similarly, the authors also emphasized the issue of high fragmentation of spot markets for digital currencies due to their unregulated nature. Therefore, the absence of such mechanisms creates the opportunities for arbitrageurs to trade across different markets. Comparatively, Ref. [18] investigated arbitrage opportunities between bitcoin cash and future markets. The authors determined that although arbitrage opportunities prevailed between December 2017 and February 2018, such opportunities faded away thereafter. The long-term evolution of arbitrage opportunities in the bitcoin market based on fine-grained data have been explored in the study [19]. Specifically, the authors have determined interesting facts. For example, the extremes in arbitrage spreads between exchanges seemed to narrow over time, but the average bitcoin spreads consistently expanded in magnitude and stability from 2013 to 2018. Next, the multiple patterns in spreads among exchanges have been identified, which can be exploited by arbitrageurs. Moreover, empirical evidence suggests that spreads increase during the early hours of a day (according to UTC), when new exchanges enter markets, and then seems to decrease in the late hours. This leads to the conclusion that arbitrage spreads expand when predominant activity is expected in the Asia-Pacific region and Europe, and narrow when trading activity moves to the American region. The authors estimated that profitable net arbitrage opportunities have been observed during the entirety of 2017 and the first quarter of 2018 each, with the estimated profit of at least $380 million that digital money has not realized its potential.

In this paper, we first explore the arbitrage opportunities over a 2-year period since 2018. The transaction-level data coming from 20 crypto-exchanges are analyzed. The diversity of crypto-currencies included in the data set highly depends on the list of trading currencies available in the exchange. Using these data, we demonstrate existing arbitrage opportunities for different crypto-currencies estimated on an individual crypto-exchange. Next, we address the question of how to measure these opportunities. Accordingly, we design our research in a way to identify dominant exchanges in the market. We rely on graph-theory technique that is used to describe network size, connectivity, and density. Then, graph centrality metrics are considered to indicate the importance of exchange in the network, which allows us to evaluate the topological positions of individual crypto-exchanges. Particularly, we introduce a new index—arbitrage ratio, based on which exchanges are ranked, providing us a detailed landscape of a crypto-market that should help us to identify the best place to buy cryptos and then to sell them. The choice of graph theory technique is first motivated by its straightforward adoption to describe a crypto-market as the network of crypto-exchanges, with possible flows of arbitrage between them [20,21,22]. Additionally, this technique allows us estimate the key roles of exchanges in generating arbitrage opportunities by measuring their topological position in a graph, which is the main goal of this study. Finally, by employing canonical correlation analysis we provide insights into what relevant information from the crypto-market might point out to profitable arbitrage opportunities.

The rest of the paper is structured as follows. Section 2 reviews the relevant works that show the evidence of market features leading to the arbitrage formation in the crypto-currency market. Section 3 presents the methodology employed for this study. First, the arbitrage definition is formulated and a new index—arbitrage ratio is proposed to arrange crypto-exchanges according to their role in generating arbitrage. Then, we explain how the crypto-network is designed from the arbitrage flows observed in the crypto-currency market using graph theory technique. Having defined a crypto-network, we additionally select network topological measures suitable to determine the role of crypto-exchange in the market. Later, we shortly outline the concept of canonical correlation analysis. The results of the study are presented in Section 4, which begins with a summary of data used in the analysis (see Section 4.1) and demonstrates the arbitrage flows observed between the pairs of crypto-exchanges. Then, the key players—crypto-exchanges that contribute to arbitrage opportunities are identified in Section 4.3. The interesting question is addressed in Section 4.4 to investigate what happens to the crypto-network if we exclude Bitcoin from the analysis. Finally, we try to identify what movements in the crypto-market signal arbitrage opportunities by means of canonical correlation. Section 5 considers the roles of crypto-exchanges determined in the study and presents future works based on the findings of this study.

## 2. Related Works

The likelihood of arbitrage opportunities is highly related to the information flow among crypto-currency exchanges, more specifically, cross-market mean and volatility spillovers. Understanding the connections between different markets is fundamental for portfolio diversification, hedging, risk management, and arbitrage purposes [23]. Moreover, the transmission of spillovers between markets is generally seen as a result of increased integration, and also because of the presence of financial contagion or systemic risk [24,25]. Within such a research area, a relevant and growing body of literature has investigated the connectedness among different crypto-exchange markets from different perspectives. For example, the realized volatility connectedness among Bitcoin exchange markets has been examined using Diebold and Yilmaz’s framework [26,27,28], which helped to identify the key exchanges that substantially contribute to the volatility formation among Bitcoin prices within the considered system [29]. In fact, Coinbase has been estimated as a top leader of the market, while Binance has been ranked unexpectedly weak. The same methodology has been applied to investigate connectedness via return and volatility spillovers across six large crypto-currencies [30]. Their results revealed that Bitcoin and Litecoin are taking dominant positions in the connected network of returns, and moreover they are the most influential ones regarding volatility spillovers. Additionally, it has been argued by [31] that the return and volatility spillover effects tend to reinforce each other, particularly during times of stress. In this regard, the authors of [32] explored regime specific spillover across crypto-currencies and the role of COVID-19 by applying a Markov regime-switching vector autoregressive model with exogenous variables. Their findings show the evidence of greater spillovers in the high volatility regime during the pandemic outbreak, which is in line with the notion of the financial contagion spread during turbulent periods.

By employing a long-memory approach, the evolution of informational efficiency and its influence on cross-market arbitrage opportunities could be estimated. In this context, the authors of [33] studied the evolution of informational efficiency in five major Bitcoin markets and its impact on cross-market arbitrage. Their results reveal that all considered markets have been close to full informational efficiency over the sample period, however the degree of market efficiency varied across markets and over time. Specifically, the findings show that the degree of estimated efficiency in the US and Australia exhibits negative effects on the cross-market arbitrage potential, whereas the efficiency degree in Canada, Europe, and the UK imposes positive impacts on arbitrage opportunities. Comparatively, the persistence in the level and volatility of Bitcoin price has been also explored [34], where the authors emphasized the importance of accounting for the impact of structural breaks during estimation procedure. This evidence is supported by [35], where the cross-sectional dependence in panels for 31 of the top market-cap crypto-currencies is examined. The panel unit root tests are utilized to allow for any cross-sectional dependence and include possible structural breaks in the panels to jointly explore the efficiency of crypto-currencies. The evidence indicate that top-ranked crypto-currencies are not efficient, which is in agreement with other published results. In the same context, the multifractality, long-memory process, and efficiency hypothesis of six major crypto-currencies (Bitcoin, Ethereum, Monero, Dash, Litecoin, and Ripple) have been explored in the study [36] using the time-rolling multifractal detrended fluctuation analysis approach. The authors concluded that the inefficiency of crypto-currency markets is time-varying, with a strong evidence of long-memory property and multifractality. The same approach has been applied in the study [37] to examine the asymmetric efficiency of crypto-currencies such as Bitcoin, Ethereum, Litecoin, and Ripple using using hourly data. The evidence shows that growth trends demonstrate stronger multifractality than downtrend. Moreover, the authors of this study asserted that the COVID-19 outbreak adversely affected the efficiency of the four crypto-currencies, with the hardest hit on Bitcoin and Ethereum, which reveals a considerable increase of inefficiency during the pandemic period. The findings support previous results that crypto-market efficiency is dynamic, and moreover, catastrophic events may enhance adverse effects on the efficiency of crypto-currencies. Additionally, it was also found that higher liquidity improves the efficiency of crypto-currencies, however higher volatility weakens it. These results are consistent with other studies [38,39,40] that highlighted the significant relation of market efficiency of crypto-currencies, with a question of liquidity and volatility.

With emergence of crypto-markets, the paradigm of traditional financial economics hardly can be applicable in the research though. In fact, the inefficiencies of rapid price changes and volatility, connectedness, different degree of access of information, unpredictable behavior of investors, and different trading tools and techniques are currently attributed to crypto-markets. All this creates favorable conditions for arbitrage development in the market.

## 3. Methodology

### 3.1. Arbitrage Definition

Arbitrage involves taking advantage of going long within the exchange where crypto-currency is underpriced and then taking a short position at the exchange where it is overpriced at a given point of time. Accordingly, let define $P_{i}^{c}\left(t\right)$ as the smallest price of crypto-currency *c* prevailing in the exchange *i* and $Q_{i}^{c}\left(t\right)$ is the quantity available to buy in *i* at time *t*. At the same time, *t* the greatest price at exchange *j* is $P_{j}^{c}\left(t\right)$ with quantity $Q_{i}^{c}\left(t\right)$ to sell. Then the arbitrage opportunity on crypto-currency *c* at time *t* is defined as:(1)$$D_{ij}^{c}\left(t\right)=V_{ij}^{c}\left(t\right)\left(P_{j}^{c}\left(t\right)\left(1-fee_{j}^{c}\right)-P_{i}^{c}\left(t\right)\left(1-fee_{i}^{c}\right)\right)-C_{ij}^{c},$$
where i,j=1,…,N, j≠i is a crypto-exchange index, *N* is a total number of exchanges available, $V_{ij}^{c}\left(t\right)=\min\left(Q_{i}^{c}\left(t\right),Q_{j}^{c}\left(t\right)\right)$ is the volume of arbitrage on currency *c* between exchanges *i*, and *j*, $Q_{i}^{c}\left(t\right)$, and $Q_{j}^{c}\left(t\right)$ are bid/ask volumes of crypto-currency *c* available in the corresponding exchange *i* or j≠i at time *t*, $fee_{j}^{c}$ is a fee charged for sell transaction, $fee_{i}^{c}$ is a fee charged for the purchase transaction in a corresponding exchange, and $C_{ij}^{c}$ is a transfer charge between exchange *i* and *j* or withdrawal fees. Henceforth, the term “taxes” will be used to consider joint exchange fees and transaction commissions.

In this paper we analyze two cases of taxes applied:There are no taxes, i.e., $fee_{i}^{c}=0$ and $C_{ij}^{c}=0$, for all crypto-currencies *c* and all exchanges i,j=1,…,N;Taxes are taken into account, i.e., $fee_{i}^{c}>0$ and $C_{ij}^{c}>0$, for all crypto-currencies *c* and all exchanges i,j=1,…,N.

The first case is important to show the overall magnitude of arbitrage possible in the market, while the second case shows potential net profit from arbitrage in the crypto-currency market. The arbitrage for the first case is strictly positive, while in the second case it may become negative due to a lack of balance between profitable quantity, price difference, and fees. In the second case we analyze only those arbitrage cases that are profitable after all taxes.

Having a complete list of exchanges and all available flows of arbitrage between exchanges, we introduce a new index—arbitrage ratio, which is used to rank the crypto-exchanges. Such ranking allows us to have a detailed landscape of the crypto-market and identify the best place to buy cryptos and then to sell them. Accordingly, we introduce the roles for crypto-exchanges such as “Buyer” and “Seller” denoting them as $D_{i}^{b,T}$ and $D_{i}^{s,T}$, where *T* denotes period (see Equations (Equation 3) and (Equation 4)). Then, the arbitrage ratio $R_{i}^{c,T}$ at exchange *i* on crypto-currency *c* (this index is not represented in some formulas below, however we understand it by default) over some predefined period *T* is defined as:(2)$$R_{i}^{c,T}=\frac{D_{i}^{b,T}-D_{i}^{s,T}}{D_{i}^{b,T}+D_{i}^{s,T}};$$
where $D_{i}^{b,T}$ is the overall arbitrage turnover from trading activities of purchase at exchange *i* on crypto-currency *c* over period *T*:(3)$$D_{i}^{b,T}=\sum_{t\in T}\sum_{m=1}^{N}\max(D_{im}^{c}\left(t\right),0)$$
and $D_{i}^{s,T}$ is the overall sales turnover at exchange *i* on crypto-currency *c* over period *T*:(4)$$D_{i}^{s,T}=\sum_{t\in T}\sum_{m=1}^{N}\max(D_{mi}^{c}\left(t\right),0).$$
The proposed ratio $R_{i}^{c,T}$ is equal to −1 if the crypto-exchange *i* is only used to purchase cryptos and arbitrage is closed somewhere else. Furthermore, the ratio $R_{i}^{c,T}$ is equal to −1 if the exchange *i* is used only to close arbitrage, i.e., to sell everything that has been bought in other exchange. All other values of ratio $R_{i}^{c,T}$ range in (−1;1) and represent a portion of arbitrage in a particular exchange *i* used for purchasing crypto-asset (positive) and a portion of arbitrage used for closing it (negative). To avoid misinterpretation of results when taxes are greater than 0 and turnover $D_{ij}^{c}\left(t\right)$ may become negative, in Equations (Equation 3) and (Equation 4) we must take only profitable transactions.

However, the arbitrage ratio does not indicate a magnitude of potential arbitrage. To solve this issue, we introduce network topology characteristics presented in the next section.

### 3.2. Network Model of Arbitrage Flows

There exist numerous crypto-currencies and a variety of markets on which they are exchanged. Arbitrage opportunities arise among these markets, therefore it can be formalized as the network model using a graph-theoretic approach [41]. Basically, a graph G=(V,E) is a connection of vertices *V* joined by edges *E*. For a crypto-currency market, we consider a directed weighted graph, where each vertex represents the crypto-exchange, and each edge defines the arbitrage flow between Buyer and Seller at some point of time *t* (Figure 1). Notably, there exist a number of edges that enter a particular vertex and a number of edges that exit this vertex. Typically, these connections existing in the graph are summarized in the adjacency matrix *A*: $A=\{A_{ij}\}$, where $a_{ij}\neq 0$, i,j=1,…,N.

Formally, using the variables introduced in Section 3.1, the edges *E* of graph *G* are the estimates of arbitrage flows at time *t*. Thus, we define the elements of adjacency matrix *A* in two ways relevant in further analysis: $A_{ij}$ represent the estimates of aggregated values of arbitrage flows either summing over time *t* or summing values over crypto-currencies *c*.

Outside the description of network by the number of vertices and edges, a broad range of network measures have been introduced to characterize graphs. Some of these measures are proposed to describe the network structure itself. In particular, edge density explains how many edges between vertices (crypto-exchanges) exist compared to how many edges between vertices are possible. Comparatively, reciprocity is a measure of the likelihood of vertices in a directed network to be mutually linked pointing in opposite directions.

In order to quantify the role of crypto-exchange in the network, the vertex centrality measures are considered. However, there exist several different measures of centrality that estimate the vertex’s importance due to their topological position in a graph. Therefore, we preferred such measures that could help us reveal the role of crypto-exchange and discover the arbitrage opportunities in the network:Hubs and authorities are used to determine the relevance of crypto-exchange in the network [42]. A good hub represents a crypto-exchange having a terminal Buyer role (that points to many other exchanges), and a good authority represents a crypto-exchange having a terminal Seller role (that is linked by many different exchanges). The authority scores of the vertices are defined as the principal eigenvector of $A^{T}A$, while the hub scores of the vertices are defined as the principal eigenvector of $AA^{T}$. Recall that matrix *A* defines the adjacency matrix. The value of scores ranges between 0 to 1, where a larger value shows the higher importance of the crypto-exchange as a Seller or Buyer, respectively;PageRank centrality was introduced by the founders of Google to rank web-pages in search engine results. It is a variant of eigencentrality [43], but the importance of a vertex (crypto-exchange) is determined through the number of edges it receives, as well as the edge propensity and the centrality of its neighbors [44]. In mathematical terms, the Pagerank centrality is defined by:
(5)$$x_{i}=\alpha \sum_{j}A_{ij}\frac{x_{j}}{k_{j}^{out}}+\beta ;$$
where $A_{ij}$ is an element of the adjacency matrix and $x_{j}$ is the element of eigenvector of matrix *A* such as Ax=λx. The parameter $k_{j}^{out}$ is the out-degree of vertex *j*, which is set to one for zero out-degree vertices to avoid division by zero, α,β>0. So, with PageRank centrality, we aim to uncover important crypto-exchanges whose accessibility goes beyond just their direct connections;Strength centrality is defined as the sum of edge values of the adjacent edges from each vertex [45]. For the network of crypto-exchanges, it describes overall arbitrage turnover of crypto-currency that has occurred in this exchange;Diversity measure shows a vertex’s connections to communities outside of its own community. Specifically, the vertex with many connections to other communities will have a higher diversity value [46]. Mathematically, the diversity of a vertex is defined as the Shannon entropy of the edge value (weights) of its incident edges:
(6)$$D_{i}=-\sum \left(p_{ij}\log\left(p_{ij}\right),j=1\ldots k_{i}\right)/\log\left(k_{i}\right);$$
where $p_{ij}=w_{ij}/\sum \left(w_{im},m=1\ldots k_{i}\right)$, $k_{i}$ is the degree of vertex *i*, and $w_{ij}$ is the weight of the edge(s) between vertices *i* and *j*. For the crypto-currency market, the crypto-exchange with diversity close to 1 has greater between-community connectivity, while value close to 0 suggest greater within-community connectivity;Betweenness centrality can be understood as a probability of crypto-exchange to occur on a randomly chosen shortest path between two crypto-exchanges [47]. In case of application, crypto-exchanges with high betweenness centrality may have a considerable influence within a network by virtue of their role over flows passing between others. Formally, the betweenness of vertex *i* is defined as follows:
(7)$$B_{i}=\sum_{j\neq k\neq i}\frac{n_{jk}^{i}}{n_{jk}}$$
where $n_{jk}$ is the number of shortest paths from *j* to *k*, and $n_{jk}^{i}$ is the number of those paths that pass through vertex *i*.

### 3.3. Canonical Correlation Analysis

Canonical correlation analysis is a multivariate statistical technique to study correlation between two sets of variables [48,49]. Suppose we have *p* variables in one set $X={\left(X_{1},\ldots ,X_{p}\right)}^{\prime }$ and *q* variables in another set $Y={\left(Y_{1},\ldots ,Y_{q}\right)}^{\prime }$. Then observed variables in both sets are linearly combined to form a set of canonical variates *U* and *V* defined as:U=A·XandV=B·Y,
where coefficients in matrices $A_{p\times p}$ and $B_{p\times q}$ are selected to maximize the canonical correlation, p≤q. This idea is summarized in Figure 2.

We have canonical variates $U=(U_{1},U_{2},\ldots ,U_{p})$ and $V=(V_{1},V_{2},\ldots ,V_{p})$, which are also known as a canonical functions. For example, $U_{1}$ is a linear combination of *p* variables from a set *X* and $V_{1}$ is a linear combination of *q* variables from *Y*. Typically, all resulting pairs of canonical variates $\left(U_{i},V_{i}\right)$ are arranged from the largest canonical correlation achieved. The number of canonical functions to be evaluated is equal to the number of variables in the smaller set. Squared canonical correlation defines the proportion of variance shared by observed variable *X* or *Y* with newly derived canonical variates $U_{i}$ and $V_{i}$, and then is used to describe the relative importance of observed variables.

## 4. Results

This section presents the relevant data used in the analysis and gives insights on arbitrage formation among different exchanges in the crypto-market. Then, the network obtained from historical arbitrage flows is investigated by means of graph theory. In the analysis, we aim to identify those crypto-exchanges where the arbitrage could have been initiated and then closed in order to earn a profit. Special attention is payed to the impact of transaction and withdrawal fees to the arbitrage network structure and crypto-exchange roles. Finally, we collect variables that represent crypto-market movements, which might indicate arbitrage opportunities depending on the situation in financial markets. Special attention is paid to the question of what happens to the network if we exclude the largest crypto-currency Bitcoin.

### 4.1. Data

The analysis covers the period 12 February 2018–30 March 2020, during which the potential arbitrage was estimated for 20 crypto-exchanges established in different countries (see Table 1). All mentioned crypto-exchanges were monitored and real-time bid/ask transactions registered for every crypto-currency analyzed. Such transaction level data were filtered to estimate potential and after-tax arbitrage using Formula (Equation 1). The resulted data set includes 62,102,537 arbitrage historical observations, among which 29,514,859 are profitable ones after fees. Notably, the country of exchange operation might influence the fiat currency used as a base currency. For comparison purposes, the total monetary amounts are provided in Euros.

Table 1 outlines the countries and names of exchanges that are included in the analysis. Additionally, the opening date and the ranking position are provided (more details may be found on particular web pages that are given in Appendix A). The last two columns indicate the potential pre-tax arbitrage for every exchange based on its role: Buyer and Seller as described in Section 3.1. As seen from Table 1, the total potential arbitrage exceeds €631.753M during the considered period. On average, each transaction could generate more than €10 of arbitrage. In further analysis, Bitfinex has been removed from the list due to very small arbitrage observed. Five crypto-exchanges were closed before the beginning of 2021. However, they together generated over €378.063M of potential arbitrage and therefore have not been removed from the following analysis. The next table provides details of the crypto-currencies that have been actually traded in the considered exchanges.

As can be seen in Table 2, the potential arbitrage has been estimated in six different crypto-currencies during the considered period. As one might expected, the greatest potential arbitrage of €588.3M comes from Bitcoin, which accounted for over 93% of all arbitrage observed. Notably, BCHEUR, BTCPLN, LTCEUR, and XLMEUR resulted comparatively in low potential arbitrage, and therefore have been removed from further analysis.

Next, we will focus on deeper analysis to estimate the arbitrage generated in every crypto-exchange individually, and then to reveal the most profitable pairs of exchanges.

### 4.2. Pairwise Analysis of Crypto-Exchanges

Arbitrage opportunities appear by taking advantage of a price difference between at least two crypto-exchanges. Let us recall the role of Buyer and Seller given to every crypto-exchange (see Section 3.1). In particular, Buyer defines the exchange where the crypto-assets are purchased, while Seller points to the terminal exchange where the assets are sold or cashed.

These flows are summarized in Figure 3, which demonstrates historically observed arbitrage if a crypto-asset is purchased in Buyer crypto-exchange *i* and then sold in Seller crypto-exchange *j*.

Figure 4 shows that historically the highest arbitrage of €112.277M was generated in Kraken, where the crypto-assets were purchased. The second largest potential profit from arbitrage, exceeding €92M, is determined in Coinfloor. Switching the roles, one can observe that DSX is the exchange where the selling of crypto-assets generated arbitrage above €220M, which is followed by EXMO that resulted in arbitrage of around €170M. There are seven exchanges (BitBay, Bitlish, Bitstamp, CEX.IO, Coindeal, Coinfloor, and Kraken) that generated arbitrage for buying over €3M. However, there existed only five exchanges (Bitlish, CEX.IO, Coindeal, DSX, and EXMO), where the total amount for closing arbitrage is over €30M. The latter two potentially generated close to €390M of arbitrage (over 60% of all potential arbitrage) during the considered period. This indicates that the market is fragmented and the closing of arbitrage is dominated by two players, i.e., DSX and EXMO crypto-exchanges.

From Figure 3 we can see that the maximal aggregated arbitrage was observed between Coinfloor (Buyer) and DSX (Seller) crypto-exchanges. Apparently, some of the arbitrage opportunities disappear after taking into account exchange fees and transaction commissions, which is jointly named as taxes herein. Therefore, in further analysis we investigate the impact of taxes on earnings from arbitrage. To do so, we consider actually observed values for taxes introduced in Equation (Equation 1) (see Section 3.1). Comparatively, the differences in the proportions of aggregated arbitrage flows are hardly visible, while the amounts of potential arbitrage have decreased after taxes are paid (see Figure 3).

To reveal the actual role of crypto-exchange that has been historically dominating, Figure 4 reports the potential arbitrage depending on the role of exchange for different crypto-currencies.

Let us discuss the share of crypto-currencies that are traded in the crypto-market. As seen from Figure 4 and Table 2, the greatest arbitrage is observed in Bitcoin (in respect to EUR, USD, and GBP). The share differs depending on the crypto-exchange. For instance, Coinfloor has similar proportions for BTCEUR and BTCGBP (exceeds €40M in Buyer role), while BTCGBP arbitrage is not significant in Bitlish at all. At the same time, Bitlish has the largest share of ETHEUR in the role of a Buyer and nearly all arbitrage opportunities on trading ETHEUR have been closed in EXMO. BTCUSD arbitrage is mainly initiated in Kraken (Buyer role) and closed in EXMO and DSX (Seller role). The two latter crypto-exchanges in the Buyer role are dominated by BTCUSD arbitrage.

Figure 5 shows the after-tax arbitrage generated depending on the role of crypto-exchange for every considered crypto-currency.

From Figure 5 it is clear that buying in Kraken generated a profit of €76.3M, followed by Coinfloor with €74.5M. Furthermore, DSX is the crypto-exchange where selling generates a profit of €177.4M, followed by EXMO with €143.4M. There existed six exchanges (Bitlish, Bitstamp, CEX.IO, Coindeal, Coinfloor, and Kraken) that generated arbitrage for buying over €30M. However, there existed only three exchanges, i.e., CEX.IO, DSX, and EXMO, where the total amount for closing arbitrage resulted in over €30M. Comparatively, DSX and EXMO generated arbitrage over €320.8M after taxes during the considered period.

Summarizing insights from Figure 3, Figure 4 and Figure 5, it is clear that, despite the taxes considered, DSX and EXMO exchanges are the best places to close arbitrage, while the best exchange to initiate arbitrage is Kraken and Coinfloor. The greatest arbitrage of €70.8M (after taxes €59.6M) is possible when buying in Coinfloor and selling in DSX crypto-exchanges. Such profit was generated by 408,974 buy/sell operations observed during the period analyzed.

Let us take a look at the average rate between potential arbitrage after- and before- taxes by taking the role of Buyer and Seller (see Table 3). Such an average rate indicates the impact of fees to the profit from arbitrage in each crypto-exchange. More specifically, the numerical value shows, on average, what portion of arbitrage is profitable. The greater the value is, the smaller the taxes paid in the particular exchange. It ranges in [0,1], where value 1 indicates that profit after taxes is the same as before taxes, while value 0 shows that there is no net profit. The latter case was observed only in three pairs DSX/Coinfloor, CoinMate/Coinroom, and SingularityX/TheRock (here Buyer/Seller roles are indicated). Additionally, from Table 3 we can see that, on average, the greatest net profit in the role of Buyer was observed in Bitsane (79%), Bitmarketlt (63%, very small turnover indeed), and Coinroom (62%), while in the role of Seller the exchanges such as EXMO (85%) and DSX (73%) were leaders.

### 4.3. Crypto-Network Topology Analysis

The question to be addressed is how the crypto-exchanges contributed to the pre-tax arbitrage observed over the considered time period. The edge density of 0.9415 indicates the high connectivity of crypto-exchanges, which means that arbitrage has been observed almost among all possible pairs of exchanges at least once throughout the period. Comparatively, the reciprocity is determined equal to 0.9814, which implies that roughly 98% of crypto-exchanges have mutually interchanged possible arbitrage flows.

Next, we look at network characteristics to be estimated for every crypto-exchange individually in the network (see Table 4). High values of hub- and authority-scores point out to the most influential crypto-exchanges across the network, connecting various pairs and carrying significant arbitrage flows. Among them, we can distinguish top hub-Buyer Coinfloor, which is followed by Kraken, Coindeal, and Bitstamp which achieved significantly lower scores. Crypto-exchanges such as DSX, EXMO, CEX.IO, and Bitlish are the big players receiing flows into their accounts and act as Sellers in the market. Comparatively, top authorities typically have a large in-degree, while top hubs possess a large out-degree. Many connections mean that the differences of crypto-currency prices are observed for many pairs of exchanges. It is equally important to have diverse “neighbors”, i.e., multiple and diverse sources imply higher probability for the arbitrage to appear. From Table 4, one can observe that comparatively, the highest diversity is determined for Coindeal, CEX.IO, and EXMO, while the lowest diversity determined for SingularityX and Coinfloor shows the existence of local community for arbitrage flows. Pagerank examines the network structure and estimate a crucial influence on the entire system. In our case, it means that crypoexchanges such as EXMO, DSX, and CEX.IO are of central importance to observe the arbitrage in the crypto-world. Correspondingly, arbitrage ratio indicates that the best exchanges to initiate arbitrage are Gatecoin, Coinfloor, and Bitsane (ratio is above +0.8), however, they have not enough strength to “feed” demand of EXMO and DSX that are the greatest Sellers or terminal points (ratio is below −0.8) of arbitrage. To fulfil the demand, Kraken, Coindeal, and even CEX.IO should be additionally considered as potential initiators of arbitrage. Comparatively, the crypto-exchanges with a high value of betweenness act as an intermediate between other exchanges in the network. It can be seen that seven out of 19 crypto-exchanges serve a link among other paired nodes for arbitrage flows, where the highest score of betweenness belongs to SingularityX. Finally, in monetary terms, the strength measure shows the arbitrage turnover observed in the crypto-exchange.

The network of crypto-exchanges with estimated pre-tax and after-tax arbitrage flows between them are correspondingly illustrated in Figure 6. The direction of arrows shows the flow of arbitrage from crypto-exchange where a crypto-asset was bought to the crypto-exchange where it was sold, while the width of arrow represents the arbitrage in the monetary value. The color of the vertex relates to the estimated arbitrage ratio (see Equation (Equation 2)). Specifically, the more violet a vertex is, the more outgoing links the exchange has, and vice versa, the more red a vertex is, the more incoming links it has. This implies that Gatecoin, Coinfloor, and Bitsane are the exchanges where arbitrage has been initiated, while EXMO and DSX are seen as exchanges where it has been marketed.

It is interesting to note that there are no significant differences in the network if we consider before- and after-tax arbitrage opportunities just by looking at Figure 6 left and Figure 6 right, which look nearly the same. However, as it was pointed out in Table 2 and at the end of Section 4.2 that taxes have a significant impact on the magnitude of arbitrage (on average 61.42% of potential arbitrage is profitable), i.e., specifically on the weights of network links. Despite visual similarities of graphs before (Figure 6 left) and after (Figure 6 right) taxes, the arbitrage possibilities are different in few aspects.

First, the overall strength of the network is different. Before taxes strength was equal to €1263.508M while after taxes, it is reduced to €889.257M. Secondly, the decrease of edge density and reciprocity (correspondingly from 0.9415 to 0.9327 and from 0.9814 to 0.9581) indicates that a rather small portion of arbitrage opportunities become no more profitable since those edges have disappeared from the network. Overall, after estimating the same list of characteristics of after-tax arbitrage network, it has been observed that the diversity and arbitrage ratio have been mostly impacted by this change. The lower values of diversity of an after-tax network indicate that the number of diverse “neighbors” of each crypto-exchange has decreased. Regarding the arbitrage ratio, it increased from 0.0371 to 0.3131, which implies that crypto-exchanges preferred acting as Buyers rather than Sellers. Considering after-tax effect individually, the leading crypto-exchanges in terms of their estimate of arbitrage strength maintained their positions in the network. By contrast, small exchanges have been more or less affected, where the largest change is observed for IncoreX, which was a very weak Seller in a pre-tax arbitrage network and became a small Buyer in the after-tax network. To sum up, it can be said that taxes have little impact on the leading roles of the crypto-market with some changes on the market structure itself.

### 4.4. Arbitrage Opportunities Excluding Bitcoin

This section explores arbitrage opportunities by excluding BitCoin from the crypto-network model. For a comparative analysis, the same technique is applied to estimate the arbitrage flows in the network as well as the network topological characteristics.

The following figures represent the arbitrage network of ETHEUR and XRPEUR crypto-currencies (see Figure 7). During the period analyzed, there was no ETHEUR arbitrage observed in CoinFalcon, Coinfloor, CoinMate, Coinroom, and DSX crypto-exchanges. Meanwhile, the arbitrage on XRPEUR was only observed in BitBay, Bitlish, Bitmarketlt, Bitstamp, CEX.IO, Kraken, and Quoinex crypto-exchanges.

As can be seen from Figure 7, the landscape and directions of arbitrage completely changed after Bitcoin cryptocurrency was excluded. Now the largest flows on trading ETHEUR are observed for exchange pairs such as Bitlish-EXMO, Kraken-EXMO, and Bitsane-EXMO. Comparatively, for the XRPEUR case we can see a considerably lower number of exchange pairs for which the arbitrage has been historically observed. Here, the largest flows on trading XRPEUR are observed for Kraken-CEX.IO and Bitstamp-CEX.IO.

Considering the network of each crypto-currency separately, the edge density dropped from 0.9327 (see Section 4.3) to 0.8516 (ETHEUR) and 0.1169 (XRPEUR) respectively. This indicates that less crypto-exchanges participate in ETHEUR and XRPEUR arbitrage. To clarify the main differences experienced in the crypto-network, Table 5 outlines the network characteristics of crypto-exchanges.

From Table 5 we can see that the roles of crypto-exchanges differ depending on the crypto-currency. EXMO that has been estimated (see Section 4.3) as the largest Seller in the network based on the arbitrage ratio also remaining a leader in the case of ETHEUR. Comparatively, Gatecoin hit the top position as a Buyer among all exchanges (see Section 4.3), but now, in the case of ETHEUR, it became a lesser Buyer, by not performing trades on XRPEUR. The most stable position is observed for Kraken, Bitstamp, Bitmarketlt, and Quoinex that are classified as Buyers in all cases analyzed. Kraken can be classified as a strong Buyer independently on the underlying crypto-asset, while Bitstamp always remains a weak Buyer. The role of Seller is more difficult to relate, because only CEX.IO can be classified as Seller in all cases. In the case of all crypto-currencies included (see Section 4.3), CEX.IO had the greatest diversity, and the arbitrage ratio was close to 0. Such a combination of characteristics allow us to name CEX.IO as a Central player in the market, having many incoming and many outgoing links. In the case of ETHEUR, it is the third largest Seller (after EXMO and BitBay) and it is the largest Seller in case of XRPEUR.

### 4.5. Relationship between Arbitrage and Some Crypto-Market Variables

To explore arbitrage opportunities over time, the collection of variables that represent weekly arbitrage opportunities in terms of total arbitrage value, number of transactions, and average transaction value for BTCEUR, ETHEUR, and XRPEUR, have been measured. Moreover, the pre-tax and after-tax arbitrage opportunities have been also estimated. Their variation over a 2-year period is depicted in Figure 8, Figure 9 and Figure 10.

It is not surprising that the greatest potential for arbitrage is observed on trading BTCEUR. If we look at the trends of weekly aggregated arbitrage value (see Figure 8), we could see that the possible arbitrage increased rapidly in September of 2018 for all crypto-currencies considered. However, the arbitrage opportunities on trading XRPEUR dropped rather quickly, while BTCEUR and ETHEUR arbitrage recovered several times but in lower amounts. What can be clearly seen in the figure is a sudden and significant growth at the end of the observed period, which might be explained by the outbreak of the COVID-19 pandemic. Comparatively, we also estimated the weekly average transaction value, which revealed slightly different insights (see Figure 9). It can be seen that in the first half of the observed period the average transaction value of arbitrage fluctuated in a range that did not apply thereafter, except the beginning of 2020. An interesting observation is that in some periods, the after-tax average transaction value is observed to be larger than the pre-tax value. It can be explained by the weekly number of transactions estimated for opportunities of profitable arbitrage before- and after- taxes, which consequently had an effect on the increase of the average transaction value itself (see Figure 10). This phenomena is mainly observed for BTCEUR and ETHEUR, when the gap between the number of transactions before- and after- taxes increases. It can be also seen that starting from September of 2018, the pre-tax number of transactions mainly fluctuated around some level, while the after-tax number experienced comparatively significant upward and downward short-run changes.

To better understand what might be driving the arbitrage opportunities over time, we include some crypto-market variables that signal about their price change and volatility observed within s week. Therefore, weekly log-returns for dominating currencies such as BTCEUR, BTCUSD, ETHEUR, XRPEUR, USDEUR, and GBPEUR have been estimated. Additionally, CRIX, which is seen as a benchmark for the crypto-market, has been also measured [50].

Spearman correlation analysis is used to determine the relationship between pairs of crypto-market variables and arbitrage opportunities measured in different terms (see Table 6).

In Table 6, the largest values of correlations are observed for the volatilities estimated for market variables and arbitrage opportunities on trading XRP. This applies to all measures of pre-tax and after-tax arbitrage opportunities. It is rather unexpected but this suggests that the dynamics of XRPEUR in some levels is related with the arbitrage opportunities observed, since the value of the correlation coefficient is approximately 0.4. Another observation is that the correlations estimated between arbitrage measures and weekly log-returns of market variables in most cases are negative or insignificant. Therefore, we could conclude that by using Spearman correlation analysis, we were not able to determine any strong relation for the pairs of considered variables.

Therefore, the canonical correlation analysis is selected in order to reflect the overall relation between two groups of variables. More specifically, we aim to determine a relationship between a collection of variables that describe arbitrage opportunities *Y* and a collection of variables that represent crypto-market movements *X*. As such, a list of canonical correlation variables $U_{i}$ and $V_{i}$ was estimated from two sets of data, *X* and *Y*, and their correlations in decreasing order are summarized in Table 7.

Table 7 shows that canonical correlation coefficient of the first pair of canonical variables $U_{1}$ and $V_{1}$ is 0.8527 at a significance level 0.05, whereas the remaining pairs of variables are not significantly correlated. This suggests that only the first pair of canonical variables is used in the subsequent analysis. Notably, the canonical correlation coefficient 0.8527 is substantially greater than the largest Spearman correlation estimated in Table 6. This implies that the relation of crypto-market movements and arbitrage opportunities is explained not individually but as a whole.

We now perform a canonical structural analysis to measure the relation between the observed variables (*X* and *Y*) and their canonical variables ($U_{1}$ and $V_{1}$). Table 8 and Table 9 summarize the obtained results.

Table 8 shows that canonical loadings of standard deviations of crypto-market variables and canonical variable $U_{1}$ achieved positive and comparatively large values, which means that the movements in the market are well reflected in price volatilities of crypto-currencies and the CRIX index. On the contrary, the weekly log returns indicate less than medium opposite correlations, except the mean of GBPEUR. Among all cross-loadings it can be observed that the increasing volatility of market variables such as standard deviations of CRIX, BTCEUR, BTCUSD, and XRPEUR indicate highly expected arbitrage opportunities. We know consider the main drives among arbitrage measures (see Table 9).

Table 9 suggests that within a group of measures estimated for arbitrage before taxes, the largest loading is observed for the aggregated arbitrage value, which is followed by the average transaction value. Comparatively, considering after-tax arbitrage opportunities we can conclude that the same arbitrage measures obtained the largest loadings. Particularly, these are variables that drive the main information about arbitrage opportunities in the crypto-market.

## 5. Discussion

In this paper, as a point of difference from many others, we analyzed high-frequency inhomogeneous arbitrage estimates observed in 20 crypto-exchanges. Such raw data allowed us to use different aggregation levels and explore pre-tax and after-tax arbitrage opportunities. The different nature of data compared to similar research imply that results obtained may be different. Our study demonstrated the amounts of potential arbitrage observed historically, which has been fluctuating and recurring over time with periods exhibiting surprisingly large flows possible and many transactions available. By means of canonical correlation analysis, we revealed that higher volatility in the market indicates higher arbitrage opportunities, which is inline with other papers published. Moreover, it has been also determined that decreasing prices of crypto-currencies also signal about more likely arbitrage appearance in the market. Therefore, the central idea of the paper was to find out which crypto-exchanges are the leaders to generate arbitrage opportunities in the crypto-world by identifying their roles as Buyers and Sellers. Moreover, these roles have been estimated in different aspects.

The analysis of the crypto-network using graph theory technique revealed how each crypto-exchange contributed to the arbitrage generated in the network. Among them, Coinfloor is a top hub where crypto prices were typically observed as lower for many pairs of exchanges, while DSX is a top authority managing the crypto-currencies with the highest prices in the network. Comparatively, Coindeal, CEX.IO, and EXMO have been connected to many other exchanges, which allowed the generation of arbitrage opportunities by ensuring multiple sources for the arbitrage to appear. These insights are also supported by PageRank centrality, based on which EXMO, DSX, and CEX.IO played key roles in the network by being linked from other important and link-parsimonious exchanges. Only seven exchanges acted as an intermediate between other exchanges in the network, among which SingularityX and IncoreX have been estimated as top intermediaries. More importantly, the introduced arbitrage ratio allowed us to investigate the roles of Buyer and Seller jointly. It turns out that overall leader of Buyers is Gatecoin (comparatively, IncoreX for ETHEUR and Kraken for XRPEUR). The leader among Sellers is EXMO (comparatively, EXMO for ETHEUR and CEX.IO for XRPEUR). However, the arbitrage ratio does not measure magnitude of arbitrage flows and influence of the exchange compared to other exchanges. On the other hand, if our results are supplemented by strength or potential arbitrage (see Table 1), then it becomes a very versatile tool that unites the pros of authority and hub score measures. Surprisingly, the taxes included in the study have not considerably impacted the leading roles of crypto-exchanges but the potential profit and the structure of the network. Moreover, it must be noted that for top Buyers, on average, taxes variate from 41% (Gatecoin and IncoreX) up to 58% (Kraken), while for top Sellers they range from 15% (EXMO) up to 47% (CEX.IO). In general, taxes reduced profit from arbitrage by nearly 30% or in our case by €375M. This naturally rises a question: Are taxes in crypto-exchanges too big? Or generalizing, it may be reformulated as follows: Is it worth overpaying for a money transfer using crypto-currencies when regular bank transfers are nearly free of charge?

Additionally, we compared the identified key players (exchanges) with their ranking scores provided in Table 1. Surprisingly, a top Seller DSX based on authority score and top prestigious based on PageRank has been closed recently. Likewise, a top Buyer Gatecoin based on an arbitrage ratio has also been closed. Moreover, CEX.IO, which has been determined as a Central player in the market, is not even ranked at all. On the other hand, EXMO classified as top Seller based on arbitrage ratio is comparatively recognized as a good one. As such, the question for a discussion is how being a key player in generating arbitrage opportunities relates to the credibility of the exchange itself. The current paper does not reveal possible reasons and therefore could be resolved for future studies. Another interesting point to be addressed is the relation between arbitrage opportunities and traded volume, which remains unexplored for the crypto-currency market. Comparatively, so far recent findings revealed an asymmetric dependence structure at the quantiles of the joint return-volume distribution for leading crypto-currencies [51,52]. Therefore, the investigation of arbitrage-volume relation might reveal insightful information for arbitrageurs or risk management purposes.

Finally, let us discuss how Buyer and Seller leaders would change if crypto-exchanges that are currently not operating (Bitlish, Bitsane, Coinroom, DSX, and Gatecoin) would be excluded from the network considered in the paper. First, we revise the Seller’s role. It is not a surprise that after DSX was eliminated, a new leader and the only clear Seller is EXMO, whose PageRank has increased up to 0.3709 and authority score up to 1. Moreover, EXMO now is the only crypto-exchange where after-tax arbitrage remains above €110M and arbitrage ratio exceeds −0.96. All these characteristics robustly indicate that EXMO is a top Seller. However, such a position points out that this crypto-exchange might be the absorbing node (according to Markov chain theory) in the arbitrage network and, moreover, lags with prices if a reason of arbitrage is the decreasing market price of a crypto-asset. In the long term such a situation could lead to a large imbalance of flows and clients would leave this crypto-exchange. Buyer role positions have been also affected, since the important players have been excluded from the network. According to a hub score and arbitrage ratio, a top Buyer Coinfloor lost its position to Kraken. This switch is not a surprise because Coinfloor had extremely large arbitrage flows to DSX (see Figure 3). Kraken, currently recognized as one of top crypto-exchanges (by CoinMarketCap ranking, see Table 1), is also identified as a leader among Buyers in our research. Our findings could be compared to a similar study [29], where crypto-exchanges were ranked based on the daily realized volatility of Bitcoin prices. Their results revealed that Coinbase, Bitstamp, Gemini, and CEX.io are the strongest net transmitters of information, while Bitfinex, Bittrex, and Binance are not strong leaders in the system of Bitcoin exchanges. Similar findings could be obtained using network analysis tools, however, they do not explain where an investor could buy a crypto-asset and where they could sell it. Hence, we can state that the role of exchanges in the crypto-world has not been yet addressed in depth and needs further research.

Overall, this study contributes to a growing body of arbitrage in crypto-currency markets. However, this study does not provide predictions on how often and where the arbitrage is likely to appear, which might be a continuation of this work in the future. Moreover, due to a limited number of crypto-exchanges and crypto-currencies included in the study, the conclusions made can not be generalized to the whole crypto-market.

## Figures and Tables

**Figure 1 entropy-23-00455-f001:**
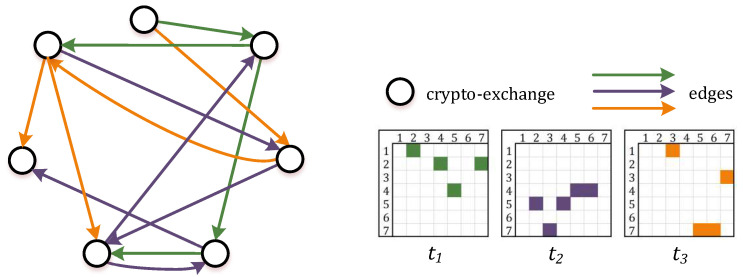
Network schematic model for arbitrage flows between crypto-exchanges.

**Figure 2 entropy-23-00455-f002:**
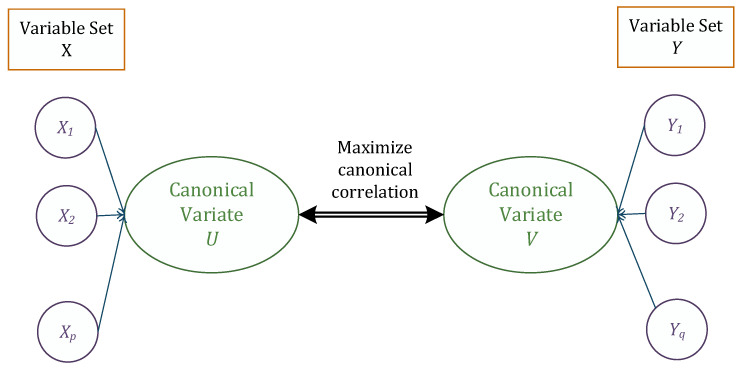
Schematic illustration of canonical correlation idea.

**Figure 3 entropy-23-00455-f003:**
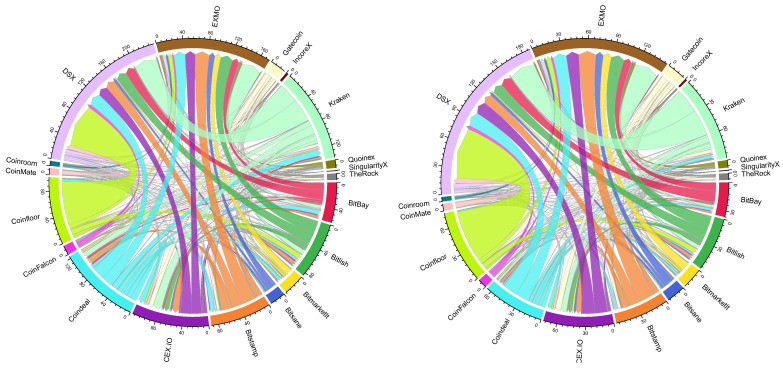
Pre-tax arbitrage (**left**) and after-tax arbitrage (**right**), €M.

**Figure 4 entropy-23-00455-f004:**
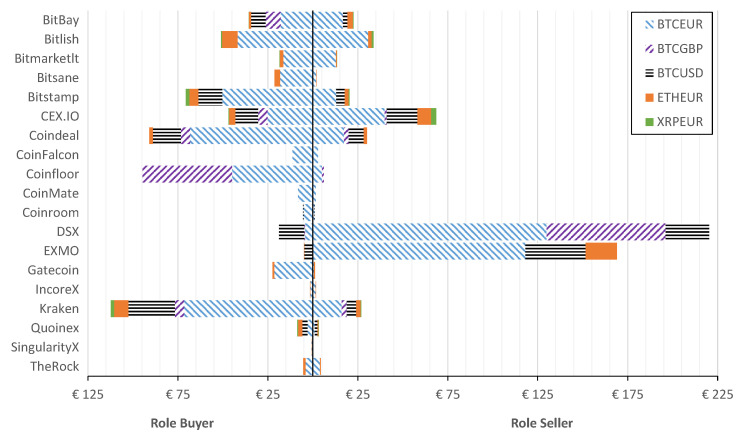
Potential pre-tax arbitrage aggregated in the exchange if the role is a Buyer or Seller.

**Figure 5 entropy-23-00455-f005:**
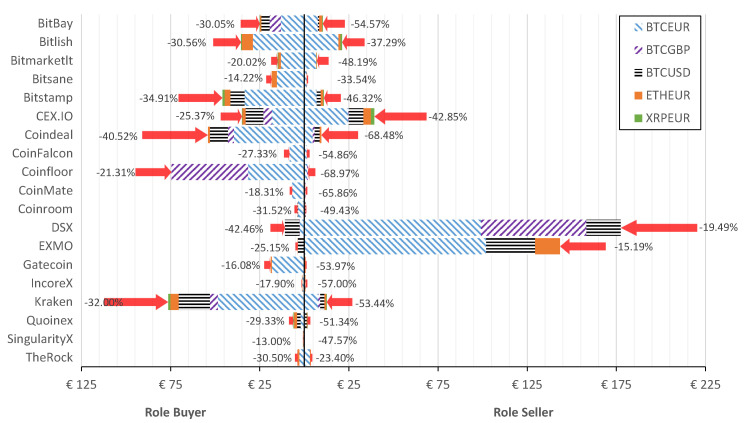
After-tax arbitrage amount after taxes in case of two roles of the exchange: Buyer and Seller. Arrows indicate the decrease (%) of arbitrage after taxes.

**Figure 6 entropy-23-00455-f006:**
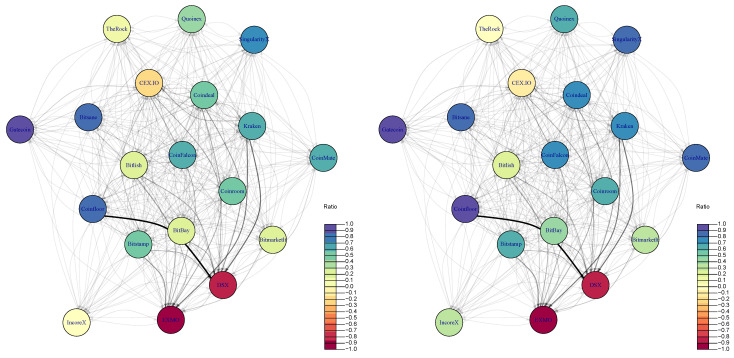
Pre-tax (**left**) and after-tax (**right**) arbitrage network.

**Figure 7 entropy-23-00455-f007:**
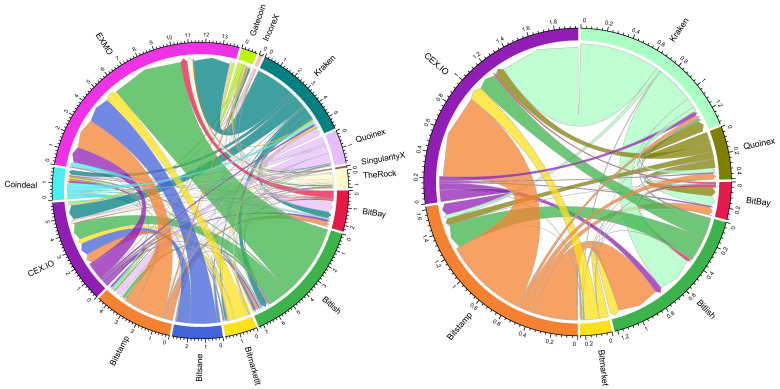
ETHEUR (**left**) and XRPEUR (**right**) arbitrage flows, €M.

**Figure 8 entropy-23-00455-f008:**
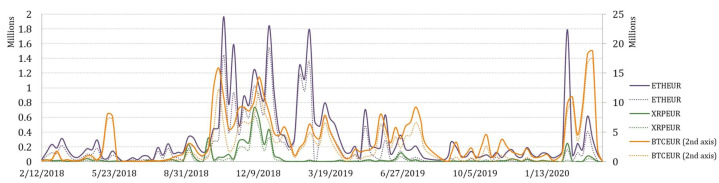
Weekly aggregated arbitrage value.

**Figure 9 entropy-23-00455-f009:**
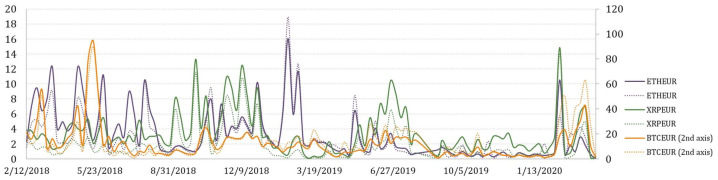
Weekly average transaction value.

**Figure 10 entropy-23-00455-f010:**
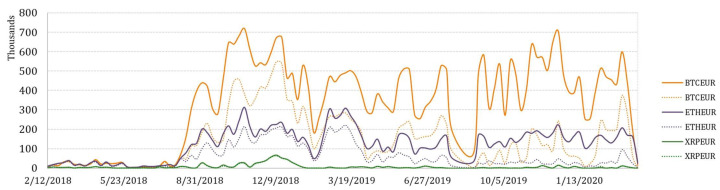
Weekly number of transactions.

**Table 1 entropy-23-00455-t001:** Crypto-currency exchanges.

Exchange	Opened	Country of HQ	Position (Ranking) *	Potential Arbitrage, €M	
				Buyer	Seller
BitBay	2014	Estonia	147	35.999	23.994
Bitfinex	2012	Hong Kong	5	0.000001	0.000033
Bitlish	2015 †	UK	-	51.043	33.755
Bitmarketlt	2013	Lithuania	–	18.604	13.500
Bitsane	2016 †	Ireland	-	21.364	2.083
Bitstamp	2011	UK	7	70.566	20.393
CEX.IO	2013	Gibraltar	–	46.901	68.556
Coindeal	2018	Malta	114	92.702	30.415
CoinFalcon	2017	UK	238	11.337	2.833
Coinfloor	2012	UK	48	94.697	6.073
CoinMate	2014	Slovakia	125	8.220	1.692
Coinroom	2016 †	Poland	-	5.451	1.560
DSX	2014 †	UK	-	18.895	220.335
EXMO	2013	UK	31	4.953	169.058
Gatecoin	2015 †	Hong Kong	-	22.411	1.165
IncoreX	2018	Estonia	-	1.624	1.643
Kraken	2011	USA	4	112.277	26.907
Quoinex	2014	Japan	12	8.658	3.261
SingularityX	2018	Lithuania	-	0.782	0.096
TheRock	2011	Italy	108	5.269	4.434
Total				631.753	631.753

Notes: † the crypto-exchange was already closed on 22 January 2021; * ranks retrieved from https://coinmarketcap.com on 22 January 2021.

**Table 2 entropy-23-00455-t002:** Crypto-currencies traded in the considered exchanges.

Crypto Currency	Ticker	ICO Date (White Paper)	Capitalisation *, €B	Potential Arbitrage, €M	Potential Profit after Taxes, %
Bitcoin	BTCEUR	09.01.2009 ^1^	482.627	414.9901	69.58
	BTCGBP			73.9420	83.41
	BTCPLN			2.1793	60.72
	BTCUSD			97.1858	65.88
Bitcoin Cash	BCHEUR	01.08.2017 ^2^	6.519	0.0034	61.04
Ethereum	ETHEUR	30.07.2015 ^3^	110.691	38.2707	66.15
Litecoin	LTCEUR	07.10.2011 ^4^	7.563	0.0005	26.71
STELLAR	XLMEUR	25.02.2016 ^5^	4.825	0.0003	50.83
Ripple	XRPEUR	20.02.2018 ^6^	9.939	5.1809	68.48
Total			622.164	631.753	61.42 **

Notes. * 22 January 2021, https://coinmarketcap.com; ** average expected profit, %. ^1^
https://bitcoin.org/bitcoin.pdf; ^2^
https://bitcoincash.org/bitcoin.pdf; ^3^
https://github.com/ethereum/wiki/wiki/White-Paper; ^4^
https://icoholder.com/en/whitepaper/litecoin-token-28667; ^5^
https://www.stellar.org/papers/stellar-consensus-protocol; ^6^
https://arxiv.org/abs/1802.07242.

**Table 3 entropy-23-00455-t003:** Average rate between potential arbitrage after- and before- taxes.

Exchange	Average Rate	
	Buyer	Seller
BitBay	0.3956	0.4604
Bitlish	0.5373	0.6464
Bitmarketlt	0.6351	0.5762
Bitsane	0.7978	0.6194
Bitstamp	0.4543	0.5248
CEX.IO	0.4711	0.5300
Coindeal	0.4994	0.3749
CoinFalcon	0.5332	0.5227
Coinfloor	0.4501	0.2758
CoinMate	0.4471	0.3748
Coinroom	0.6281	0.5023
DSX	0.4315	0.7341
EXMO	0.5962	0.8490
Gatecoin	0.5899	0.4363
IncoreX	0.5845	0.3855
Kraken	0.4216	0.4285
Quoinex	0.5586	0.6126
SingularityX	0.4893	0.5792
TheRock	0.5009	0.4955

**Table 4 entropy-23-00455-t004:** Pre-tax arbitrage network characteristics for all crypto-exchanges considered.

Exchange	Diversity	Betweenness	Strength, €M	Pagerank	Authority Score	Hub Score	Arbitrage Ratio
BitBay	0.6758	0	59.993	0.0555	0.0769	0.3026	0.2001
Bitlish	0.6767	0	84.799	0.0384	0.1447	0.3896	0.2039
Bitmarketlt	0.6950	0	32.104	0.0225	0.0471	0.1025	0.1590
Bitsane	0.5164	0.0359	23.447	0.0098	0.0062	0.1408	0.8224
Bitstamp	0.6768	0	90.959	0.0694	0.0430	0.4260	0.5516
CEX.IO	0.7597	0	115.457	0.109	0.2095	0.3523	−0.1876
Coindeal	0.7831	0	123.117	0.0722	0.1058	0.4643	0.5059
CoinFalcon	0.6134	0	14.170	0.0114	0.0093	0.0944	0.6001
Coinfloor	0.3504	0.0523	100.770	0.0143	0.0193	1	0.8795
CoinMate	0.4615	0.0556	9.912	0.0094	0.0061	0.0892	0.6586
Coinroom	0.5951	0	7.011	0.0099	0.0054	0.0348	0.5549
DSX	0.6705	0	239.230	0.2186	1	0.0537	−0.8420
EXMO	0.7283	0	174.012	0.2173	0.5235	0.0263	−0.9431
Gatecoin	0.6033	0	23.576	0.0091	0.0043	0.0451	0.9011
IncoreX	0.6573	0.3856	3.267	0.0096	0.0055	0.0111	−0.0059
Kraken	0.6490	0	139.184	0.0817	0.0617	0.7884	0.6134
Quoinex	0.7009	0.2026	11.919	0.0237	0.0093	0.0351	0.4528
SingularityX	0.3227	0.9314	0.878	0.0080	0.0003	0.0047	0.7806
TheRock	0.6497	0.2026	9.703	0.0104	0.0062	0.0325	0.0860

**Table 5 entropy-23-00455-t005:** Network characteristics for all crypto-exchanges that generated arbitrage on ETHEUR and XRPEUR.

	ETHEUR							XRPEUR						
	**Diversity**	**Betweenness**	**Strength, €M**	**Page Rank**	**Authority Score**	**Hub Score**	**Arbitrage Ratio**	**Diversity**	**Betweenness**	**Strength, €M**	**Page Rank**	**Authority Score**	**Hub Score**	**Arbitrage Ratio**
BitBay	0.666	0	2.343	0.089	0.051	0.093	−0.436	0.690	0.046	0.286	0.050	0.109	0.032	−0.664
Bitlish	0.438	0.25	7.276	0.230	0.053	1	0.672	0.770	0	1.223	0.170	0.476	0.193	−0.367
Bitmarketlt	0.649	0	1.838	0.022	0.015	0.181	0.646	0.526	0.065	0.240	0.026	0.017	0.179	0.696
Bitsane	0.381	0.333	2.899	0.013	0.003	0.394	0.943							
Bitstamp	0.653	0	4.563	0.043	0.039	0.426	0.420	0.679	0	1.623	0.194	0.057	1	0.516
CEX.IO	0.756	0.071	5.996	0.089	0.281	0.207	−0.403	0.670	0	1.980	0.245	1	0.040	−0.783
Coindeal	0.880	0	1.876	0.045	0.045	0.073	0.107							
EXMO	0.646	0	13.917	0.347	1	0.001	−0.991							
Gatecoin	0.780	0	0.908	0.031	0.016	0.043	0.355							
IncoreX	0.258	0.199	0.211	0.011	0.000	0.034	0.972							
Kraken	0.644	0	5.755	0.037	0.015	0.533	0.622	0.616	0	1.341	0.074	0.032	0.940	0.730
Quoinex	0.595	0.25	1.928	0.015	0.006	0.018	0.810	0.809	0	0.404	0.035	0.062	0.102	0.511
SingularityX	0.165	0.718	0.097	0.011	0.000	0.018	0.884							
TheRock	0.704	0.269	1.023	0.017	0.009	0.082	0.545							

**Table 6 entropy-23-00455-t006:** Spearman correlations.

		Average							Standard Deviation						
		BTCEUR	BTCUSD	ETHEUR	XRPEUR	USDEUR	GBPEUR	CRIX	BTCEUR	BTCUSD	ETHEUR	XRPEUR	USDEUR	GBPEUR	CRIX
	**Pre-tax arbitrage**														
BTCEUR	Aggregated arbitrage value	−0.11	−0.11	−0.07	−0.06	−0.01	−0.06	−0.06	0.03	0.05	0.15	0.00	−0.06	0.15	0.05
	Number of transactions	−0.16	−0.16	−0.09	−0.06	−0.06	0.10	−0.12	−0.19	−0.18	−0.09	−0.19	−0.25	0.10	−0.18
	Average transaction value	−0.07	−0.07	−0.08	−0.10	0.03	−0.18	−0.04	0.42	0.42	0.36	0.34	0.28	0.10	0.37
ETHEUR	Aggregated arbitrage value	−0.02	−0.04	0.03	−0.03	0.03	−0.08	0.02	0.03	0.06	0.18	0.06	0.16	0.08	0.11
	Number of transactions	−0.09	−0.09	−0.02	−0.02	−0.04	0.12	−0.03	−0.23	−0.21	−0.07	−0.17	−0.14	0.16	−0.20
	Average transaction value	0.00	−0.02	0.02	−0.06	0.11	−0.14	−0.01	0.27	0.27	0.28	0.27	0.39	−0.04	0.30
XRPEUR	Aggregated arbitrage value	−0.03	−0.06	−0.05	−0.04	0.12	−0.08	−0.05	0.36	0.38	0.40	0.38	0.19	−0.05	0.41
	Number of transactions	−0.03	−0.06	−0.06	−0.08	0.10	−0.05	−0.08	0.24	0.27	0.28	0.25	0.15	0.00	0.30
	Average transaction value	−0.02	−0.05	−0.02	0.00	0.09	−0.15	0.00	0.38	0.38	0.42	0.39	0.24	−0.05	0.40
	**After-tax arbitrage**														
BTCEUR	Aggregated arbitrage value	−0.09	−0.09	−0.08	−0.07	0.01	−0.10	−0.05	0.10	0.12	0.22	0.07	0.03	0.17	0.14
	Number of transactions	−0.09	−0.10	−0.08	−0.07	−0.03	−0.04	−0.06	−0.06	−0.04	0.13	−0.03	0.00	0.15	0.03
	Average transaction value	−0.07	−0.06	−0.05	−0.04	0.03	−0.14	0.00	0.31	0.32	0.29	0.24	0.10	0.13	0.27
ETHEUR	Aggregated arbitrage value	−0.05	−0.08	−0.02	−0.07	0.07	−0.09	−0.02	0.07	0.10	0.26	0.18	0.27	0.07	0.18
	Number of transactions	−0.08	−0.09	−0.06	−0.06	−0.04	−0.02	−0.04	−0.12	−0.09	0.13	0.03	0.03	0.06	0.02
	Average transaction value	−0.06	−0.08	−0.02	−0.10	0.13	−0.17	−0.04	0.24	0.24	0.28	0.27	0.40	0.02	0.25
XRPEUR	Aggregated arbitrage value	−0.03	−0.06	−0.05	−0.04	0.11	−0.08	−0.06	0.38	0.40	0.41	0.38	0.19	−0.07	0.42
	Number of transactions	−0.03	−0.05	−0.06	−0.08	0.09	−0.06	−0.07	0.25	0.29	0.29	0.25	0.15	0.00	0.32
	Average transaction value	−0.02	−0.04	−0.03	0.00	0.08	−0.11	0.00	0.42	0.42	0.47	0.44	0.23	−0.10	0.44

**Table 7 entropy-23-00455-t007:** Test of H0: The canonical correlations are zero.

	Canonical	Wilks’ Lambda Test		
	Correlation	Stat	Approx	*p*-value
($U_{1}$, $V_{1}$)	0.8527	0.0150	1.5450	0.000004
($U_{2}$, $V_{2}$)	0.7178	0.0549	1.1661	0.070809
($U_{3}$, $V_{3}$)	0.6369	0.1132	0.9886	0.531021

Notes. *U_i_* is a linear combination of crypto-market variables *X*, *V*_1_ is a linear combination of arbitrage variables *Y*.

**Table 8 entropy-23-00455-t008:** Canonical structural analysis of crypto-market variables.

		$U_{1}$	$V_{1}$
		Canonical Loading	Cross Loading
Average	BTCEUR	−0.2059	−0.1756
	BTCUSD	−0.2374	−0.2025
	ETHEUR	−0.1910	−0.1628
	XRPEUR	−0.1587	−0.1353
	USDEUR	0.2758	0.2352
	GBPEUR	−0.5587	−0.4764
	CRIX	−0.2126	−0.1813
Standard deviation	BTCEUR	0.7994	0.6817
	BTCUSD	0.7794	0.6646
	ETHEUR	0.7169	0.6113
	XRPEUR	0.4607	0.3928
	USDEUR	0.6452	0.5501
	GBPEUR	0.4196	0.3578
	CRIX	0.7313	0.6236

**Table 9 entropy-23-00455-t009:** Canonical structural analysis of weekly crypto-arbitrage measures.

		$U_{1}$	$V_{1}$
		Cross Loading	Canonical Loading
	**Pre-tax arbitrage**		
BTCEUR	Aggregated arbitrage value	0.5937	0.5062
	Number of transactionss	−0.0249	−0.0213
	Average transaction value	0.5037	0.4295
ETHEUR	Aggregated arbitrage value	0.0960	0.0818
	Number of transactions	−0.0234	−0.0200
	Average transaction value	0.1636	0.1395
XRPEUR	Aggregated arbitrage value	0.2101	0.1792
	Number of transactions	0.2439	0.2080
	Average transaction value	0.2560	0.2183
	**After-tax arbitrage**		
BTCEUR	Aggregated arbitrage value	0.7147	0.6095
	Number of transactions	0.2689	0.2293
	Average transaction value	0.6025	0.5138
ETHEUR	Aggregated arbitrage value	0.0992	0.0846
	Number of transactions	0.0690	0.0589
	Average transaction value	0.0902	0.0769
XRPEUR	Aggregated arbitrage value	0.1666	0.1420
	Number of transactions	0.2543	0.2169
	Average transaction value	0.1486	0.1268

## Data Availability

Not applicable.

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
