# Peer review of "Key Roles of Crypto-Exchanges in Generating Arbitrage Opportunities"

_entropy, 2021, doi:10.3390/e23040455_

Round 1

Reviewer 1 Report

The authors of this paper investigate the arbitrage opportunities in the cryptocurrency markets based on transaction-level data covering 20 exchanges. Using a combination of methods, the authors indicate the existence of profitable arbitrage opportunities, which is important to investors and other market participants in the cryptocurrency markets.

The paper is well motivated and written. The applied methods are suitable and timely.  However, I provide some comments to enhance the quality of the analysis and better position the paper within the growing literature on the information flow among cryptocurrencies and cryptocurrency exchanges. My comments are in no particular order.

  • There is nothing wrong with the introduction section. It contains the main elements and components to motivate the topic and research question and position it within the related studies. However, it is quite long and slow. I suggest splitting it into an introduction and a litertaure review section.
  • Based on the above, I would like to see more recent papers dealing with the information flow among cryptocurrency exchanges (e.g., Realised volatility connectedness among Bitcoin exchange markets, Finance Research Letters) and papers on market efficiency of cryptocurrencies during the covid-19 (Regime specific spillover across cryptocurrencies and the role of COVID-19, Financial Innovation; Asymmetric efficiency of cryptocurrencies during COVID19. Physica A), long memory (The risks of crytocurrencies with long memory in volatility, non-normality  and  behavioural  insights, Applied Economics; Modelling Long Memory Volatility in the Bitcoin Market: Evidence of Persistence and Structural Breaks. International Journal of Finance and Economics), trading volume and cryptocurrencies return and volatility (Tail dependence in the return-volume of leading cryptocurrencies. Finance Research Letters; Can volume predict Bitcoin returns and volatility? A quantiles-based approach. Economic Modelling). Furthermore, some recent papers should be cited to motivate the choice of the network graph theory technique.
  • In Section 3.1, More explanations are needed regarding the extraction of the number of arbitrage historical observations.
  • Why are the monetary amounts given in euros? Please explain.
  • It is not clear if the profitable arbitrage opportunities are related to trading volume?
  • The authors must discuss their findings in light of prior studies (including those mentioned in comment #2). This is very useful to add more value to the empirical analyses/findings.
  • The abstract can be improved by highlighting the applied methods and the main findings.
  • Tables and figures are not very self-explanatory.

Author Response

Thank you for review it was very helpfull. Please see the attachment.

Reviewer 2 Report

Thank you very much for this opportunity to read your paper. 

I have two major concerns.

First, the paper is poorly positioned in the cryptocurrency literature, and basically ignored the majority of papers published in this field. Referencing style used seem to be not very helpful with it, since authors referred to the same papers in different places and used different numbers. I would recommend to check any systematic reviews of cryptocurrency literature field to find some key citations to clarify your contributions.

Second, presentation of the results. While network analysis is getting more popular in Finance, I am struggling to get any meaningful results from Figures 6-7.

Overall, I like the idea behind this paper but I think it can be improved. Best of luck! 

Author Response

(The authors gave the same response as above.)

Round 2

Reviewer 1 Report

The authors have carefully addressed my concerns and considered my suggestions. I recommend Accept. 

Reviewer 2 Report

I still have some minor concerns regarding the presentation of graphs, however, I find this paper quite interesting, and overall the quality of the manuscript has been improved. Therefore, I have no hesitations to recommend it for publication.